Subject Area:
cellular biology/genetics/microbiology/
molecular biology

Keywords:
condensin, chromosome segregation, fission yeast, mitosis, transcriptional termination, auxin-inducible degron

Authors for correspondence:
Norihiko Nakazawa
e-mail: nakazawa@oist.jp
Mitsuhiro Yanagida
e-mail: myanagid@gmail.com

†Present address: Kochi University of Technology, School of Environmental Science and Engineering, Life Science and Technology Course, Chromosome Function and Regulation Laboratory, 185 Miyanokuchi, Tosayamada-cho, Kami, Kochi, 782-8502, Japan.

# Condensin locates at transcriptional termination sites in mitosis, possibly releasing mitotic transcripts

Norihiko Nakazawa†, Orie Arakawa and Mitsuhiro Yanagida

Okinawa Institute of Science and Technology Graduate University, G0 Cell Unit, Onna-son, Okinawa 904-0495, Japan

NN, 0000-0001-8386-0812; MY, 0000-0003-0293-5654

Condensin is an essential component of chromosome dynamics, including mitotic chromosome condensation and segregation, DNA repair, and development. Genome-wide localization of condensin is known to correlate with transcriptional activity. The functional relationship between condensin accumulation and transcription sites remains unclear, however. By constructing the auxin-inducible degron strain of condensin, herein we demonstrate that condensin does not affect transcription itself. Instead, RNA processing at transcriptional termination appears to define condensin accumulation sites during mitosis, in the fission yeast *Schizosaccharomyces pombe*. Combining the auxin-degron strain with the *nda3* β-tubulin cold-sensitive (cs) mutant enabled us to inactivate condensin in mitotically arrested cells, without releasing the cells into anaphase. Transcriptional activation and termination were not affected by condensin's degron-mediated depletion, at heat-shock inducible genes or mitotically activated genes. On the other hand, condensin accumulation sites shifted approximately 500 bp downstream in the auxin-degron of 5′-3′ exoribonuclease Dhp1, in which transcripts became aberrantly elongated, suggesting that condensin accumulates at transcriptionally terminated DNA regions. Growth defects in mutant strains of 3′-processing ribonuclease and polyA cleavage factors were additive in condensin temperature-sensitive (ts) mutants. Considering condensin's *in vitro* activity to form double-stranded DNAs from unwound, single-stranded DNAs or DNA-RNA hybrids, condensin-mediated processing of mitotic transcripts at the 3′-end may be a prerequisite for faithful chromosome segregation.

## 1. Introduction

In proliferative cells, mitotic chromosome condensation and subsequent segregation of chromosomes to daughter cells are essential for faithful inheritance of genomic DNA. Condensin is an evolutionarily conserved hetero-pentameric protein complex that is essential for chromosome organization, not only for mitotic chromosome segregation, but also for diverse roles such as recombination and repair [1–6]. Transcription is one of the key determinants of condensin loading on chromosome DNA. Correlation between condensin association and transcription was documented for the first time in budding yeast, *Saccharomyces cerevisiae.* Chromatin immunoprecipitation (ChIP) experiments showed that condensin occupancy at ribosomal DNA (rDNA) repeats is controlled by RNA polymerase I (RNAP I)-dependent transcription [7,8]. Then, mapping of condensin binding sites at the whole-chromosome level reported that condensin directly associates with RNA polymerase III (RNAP III)-transcribed genes, such as tRNA genes, in budding and fission yeasts [9–11]. Fission yeast, *Schizosaccharomyces pombe*, condensin accumulates at genes highly transcribed by RNAP II in interphase and mitosis [12–16]. Preferential binding to active genes has also been

royalsocietypublishing.org/journal/rsob   Open Biol. 9: 190125

proposed in higher eukaryotic condensins (condensin I and II) [13,17–19]. Even in prokaryotic condensin, a parallel finding was made [20], suggesting that transcription-dependent condensin accumulation may be critical to understand its molecular activity on mitotic chromosomes.

We showed previously that *S. pombe* condensin is enriched at the 3′-end of RNAP II-transcribed mitotically upregulated and heat shock-inducible genes [12]. ChIP-seq profiles confirmed that condensin preferentially accumulates around transcriptional termination sites, rather than transcriptional start sites [13]. This binding property of condensin implies an involvement of condensin in transcriptional termination, and provides a clue to the physiological significance of condensin at actively transcribed genes. However, due to the lack of an experimental system, we could not address these questions under conditions of inactivated condensin and/or transcriptional termination factors in mitotic cells. At least in asynchronous cycling cell populations, a direct role for condensin in gene regulation has been denied in fission yeast [21]. Budding yeast condensin showed contradictory roles of condensin in transcription under cycling and quiescent conditions [22,23]. On the other hand, the functional relationship between *S. pombe* condensin and transcriptional termination at mitotically activated genes is not clear. In this study, we constructed *S. pombe* strains in which conditional degradation of condensin or 3′-end RNA processing factors occurs in mitotically arrested cells. By combining these degradation strains with β-tubulin cold-sensitive (cs) mutation, we could examine how mitotic condensin accumulation and transcriptional termination processes functionally interact.

# 2. Results

## 2.1. Construction of an auxin-inducible degron strain for the condensin Cut14/SMC2 subunit

We previously used temperature-sensitive (ts) mutant alleles of condensin subunits for functional analyses [2,24–27]. A technical problem was that condensin mutant proteins existed even at the restrictive temperature. In addition, a temperature shift-up procedure (from 20°C to 36°C) could not maintain mitotic arrest in a cold-sensitive (cs) *nda3-KM311* β-tubulin mutant strain [28]. To avoid these disadvantages of ts or cs mutant strains, we constructed an auxin-inducible degron (aid) strain of the condensin Cut14/SMC2 subunit (figure 1a) [29,31]. We tagged the *cut14*⁺ gene with the full-length target peptide of auxin (IAA17, 229-amino acids) and a haemagglutinin (HA) tag at the C-terminus, and then chromosomally integrated the *cut14-aid-2HA* fusion gene into a strain expressing skp1-AtTIR1-NLS proteins, the modified F-box protein complex that binds auxin, previously optimized in *S. pombe* [30]. Protein extracts of the resulting strain were examined by immunoblotting with antibodies against HA. Upon addition of 2 mM auxin, the protein level of Cut14-aid-2HA decreased to about 50% of that without auxin within 1 h, and to 10% in 4 h at 20°C (figure 1b). The strain expressing both Cut14-aid and skp1-AtTIR1-NLS (designated *cut14-aid* strain, hereafter) showed growth defects on auxin-containing solid media, although the strains expressing either one alone grew normally, as did the wild-type (figure 1c). In the presence of auxin, *cut14-aid* cells failed to segregate mitotic chromosomes,

showing phi-shaped chromosomes, followed by 'cut (cell untimely torn)' phenotype [32] with septum formation (figure 1d,e). This defective phenotype is indistinguishable from those of previously reported condensin ts mutant strains, at least in liquid culture condition [2,24]. This provided the experimental tool necessary to inactivate *S. pombe* condensin without a temperature shift, which enabled us to degrade condensin while keeping the cells in mitotic arrest.

## 2.2. Transcriptional induction is not affected in condensin-aid cells

We previously tested transcriptional induction of the genes encoding heat-shock protein (hsp) in *cut14-208* ts mutants in combination with the *nda3* cs mutant strain [12]. In that experiment, we could not exclude the possibility that condensin was not sufficiently inactivated, because the mRNA level of hsp genes was measured immediately after a temperature up-shift. To sufficiently inactivate condensin in order to examine whether defective condensin affects transcriptional induction in mitotically arrested cells, we used the *cut14-aid* degron mutant strain in the *nda3* cs mutant. First, *nda3* single or *nda3 cut14-aid* double-mutant strains were arrested in a prometaphase-like stage at 20°C for 4 h (figure 2a). Second, 2 mM auxin was added to the culture medium for an additional 4 h to degrade Cut14-aid proteins in the mitotically arrested condition at 20°C, and cells were shifted to 36°C for rapid induction of hsp gene expression during mitosis. Levels of *ssa1*⁺ and *hsp90*⁺ transcripts were then measured by northern blot analysis after the heat shock. Regardless of auxin addition, *nda3 cut14-aid* double-mutant cells immediately produced *ssa1*⁺ and *hsp90*⁺ transcripts after heat shock at 36°C (figure 2b). We also performed reverse transcriptional-quantitative PCR (RT-qPCR) using probes specific to *ssa1*⁺ and *hsp90*⁺ genes (electronic supplementary material, figure S1). Rapid production of hsp gene transcripts was detected in Cut14-degraded cells, to the same extent as in Cut14-intact cells. *ssa1*⁺ and *hsp90*⁺ transcripts increased about 60- and 5-fold, respectively, at 30 min (compared to 0 min). These results strongly suggest that condensin inactivation does not affect transcriptional activation during mitosis.

## 2.3. Condensin destruction does not influence transcriptional termination

Condensin preferentially binds to transcriptional termination sites of actively transcribed genes [12,13]. We examined whether transcriptional termination is affected in condensin-aid cells during mitosis. It has been reported that defects in transcriptional termination factors cause extended transcripts, called transcriptional read-through, at the 3′ end of the genes [33–37]. We also constructed aid strains for Dhp1 and Rna14, as described in the *cut14-aid* strain (electronic supplementary material, figure S2). Rna14 is an mRNA cleavage and polyadenylation factor that processes the 3′-end of RNA transcripts [34], while Dhp1 is a 5′-3′ exoribonuclease that is essential for RNA processing in transcriptional termination [35,38]. These aid strains were then crossed with *nda3* cs mutants to measure read-through mRNA products of mitotically transcribed genes (*ecm33*⁺ and *gas1*⁺) in mitotically arrested cells (figure 2c). In addition to the approximately 300 bp RT-PCR products (Fw-Rv1),

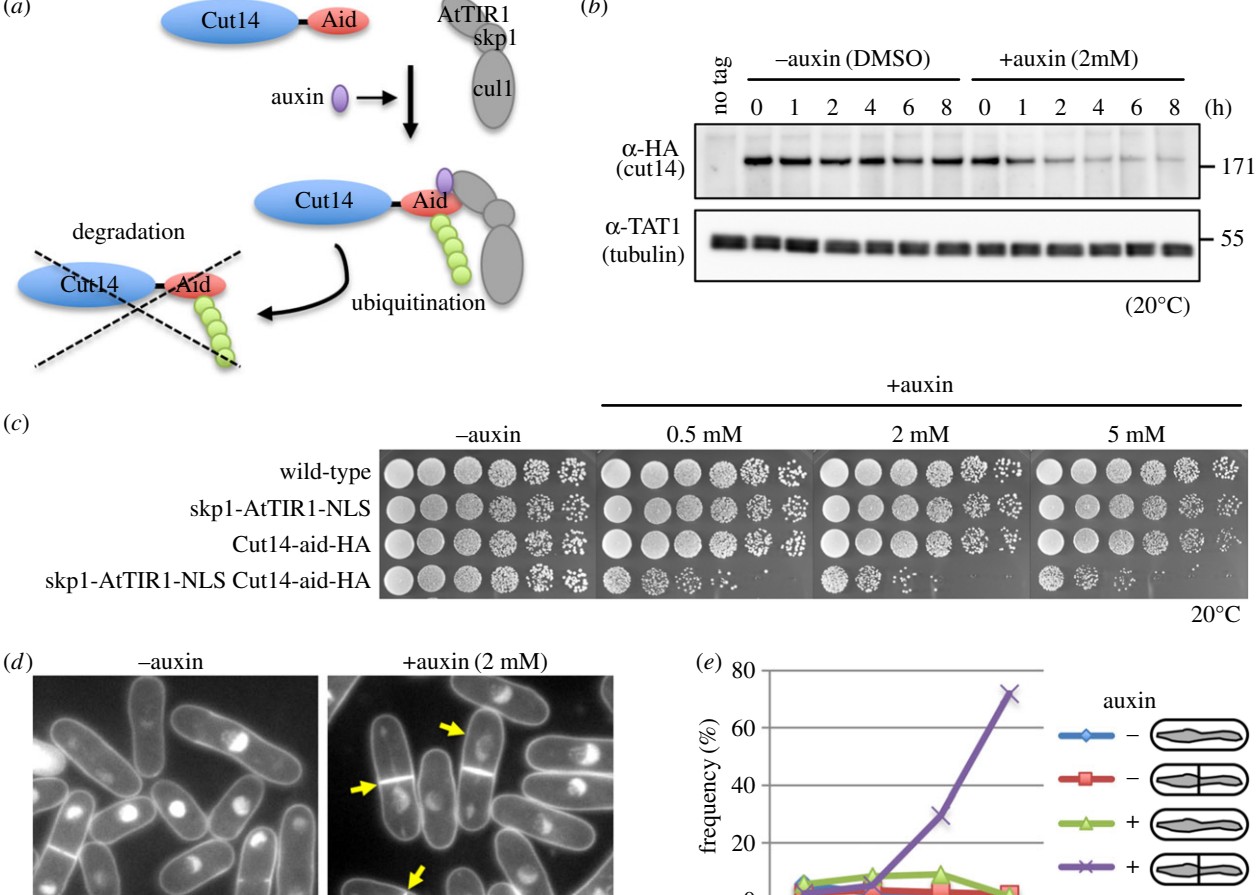

**Figure 1.** Construction of auxin-inducible degron (aid) for the condensin Cut14/SMC2 subunit. (*a*) Schematic illustration of auxin-inducible destruction [29,30] of the condensin Cut14/SMC2 subunit. (*b*) Protein level of Cut14-aid-HA in the presence or absence of auxin. Cells expressing both Cut14-aid-HA and skp1-AtTIR1-NLS proteins were cultivated at 20°C in the presence of DMSO (mock treatment) or auxin (2 mM) for indicated periods and whole cell extracts were examined by immunoblotting with anti-HA and TAT1 (tubulin) antibodies. Cells expressing only skp1-AtTIR1-NLS were used as negative controls in the presence of DMSO (no tag). (*c*) Cell growth on auxin solid media. Cells expressing either Cut14-aid-HA or skp1-AtTIR1-NLS, or both proteins were serially diluted (5×) and spotted onto YPD plates with or without auxin (0.5, 2, 5 mM) at 20°C. (*d*) DAPI-stained micrographs of cells expressing both Cut14-aid-HA and skp1-AtTIR1-NLS at 20°C for 4 h. In the presence of auxin, cells displayed phi-shaped chromosomes, followed by 'cut' phenotypes with septum formation (arrows). Scale bar, 10 μm. (*e*) Frequency of abnormal chromosome segregation in the presence or absence of auxin (2 mM). phi-shaped chromosomes with or without septation are illustrated.

approximately 500 bp products (Fw-Rv2), which appear to represent abnormally extended transcripts (figure 2*d*), were amplified in *dhp1-aid* and *rna14-aid* cells in the presence of auxin, but not in *cut14-aid* cells (figure 2*e*). Extended PCR products also were not detected in *cut14-208* ts (mutation in the coiled-coil region, S861P) and *cut14-y1* ts (mutation in the hinge region, L543S) mutants at the restrictive temperature [26,39], unlike *dis3-54* cs (exosome 3′-5′ exoribonuclease) [40], *rna14-393* ts and *cft1-665* ts (mRNA cleavage and polyadenylation specificity factors) mutant cells (electronic supplementary material, figure S3). Therefore, these data indicate that condensin is not required for transcriptional termination at mitotically transcribed genes. This result is consistent with previous transcriptomic analysis performed on asynchronously cycling condensin mutant cells [21].

## 2.4. Condensin accumulates at altered termination sites in *dhp1-aid* cells

To examine the impact of RNA-processing on condensin binding at transcriptional termination sites, we tested condensin accumulation in *dhp1-aid* cells, which produce extended transcripts. *nda3 dhp1-aid* double mutant was arrested in early mitosis for 4 h, and then Dhp1-aid proteins were degraded in the presence of 2 mM auxin for 4 h (figure 3*a*). Lengths of mRNA transcripts and condensin binding sites were subsequently analysed by 3′ RACE (rapid amplification of cDNA ends) and ChIP-qPCR (chromatin immunoprecipitation-quantitative PCR), respectively. Consistent with the read-through assay in figure 2*e*, the 3′ RACE assay detected aberrant 3′-elongated transcripts in *nda3 dhp1-aid* double mutants, which were approximately 500 bp longer than those of *nda3* single mutants (figure 3*b*), as well as the *nda3 rna14-aid* cells. In the absence of auxin (Dhp1 was intact), the ChIP-qPCR assay showed that FLAG-tagged Cut14 bound preferentially at the end of the 3′-UTR (probe 5 in figure 3*c*, top) of *ecm33+* and *gas1+* genes. In the presence of auxin, Dhp1 was degraded; however, accumulation sites for Cut14-FLAG were obviously shifted about 500 bp in the 3′ direction (probe 6 in figure 3*c*, bottom). These results strongly suggest that condensin specifically associates with the elongated 3′-end of

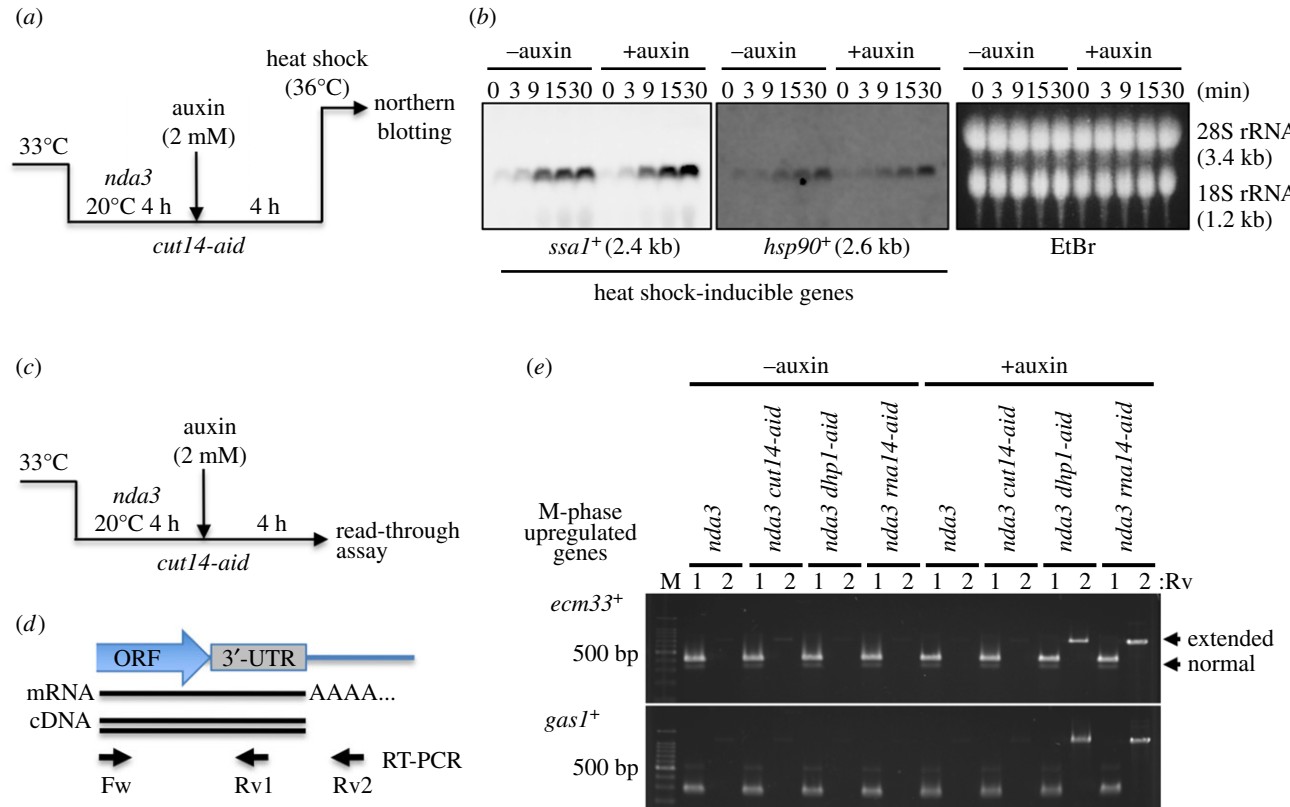

**Figure 2.** Transcriptional induction and termination were not affected in *cut14-aid* cells. (*a*) Procedures for Cut14 degradation in mitotically arrested cells and transcriptional induction of heat shock-inducible genes (hsp genes). *nda3-KM311* β-tubulin cs mutant cells expressing Cut14-aid protein were cultured at 20°C for 4 h, and incubated for an additional 4 h in the presence or absence of auxin. Cells were then shifted to 36°C (heat shock) for transcriptional induction of the hsp genes. (*b*) Northern blot analysis of the hsp genes, *ssa1*[+] and *hsp90*[+], in *cut14-aid* strains. Cells were harvested at the indicated times after heat shock in the presence or absence of auxin. Specific probes to detect these RNA products were used. Ethidium bromide (EtBr) staining confirmed equal loading by detecting 28S and 18S rRNAs. (*c*) Procedures for Cut14 degradation in mitotically arrested cells and the transcriptional read-through assay. (*d*) Schematic of the read-through assay. Total RNA was extracted, and then cDNA was reverse-transcribed by RT-PCR. Abnormally extended RNA product was amplified using the Fw-Rv2 primer set, in addition to PCR products with the Fw-Rv1 set. (*e*) Electrophoretic separation of RT-PCR products from *nda3* mutant cells expressing Cut14-aid protein in the presence or absence of auxin. *nda3* mutant cells expressing Dhp1 (5′-3′ exoribonuclease)-aid or Rna14 (mRNA cleavage and polyadenylation factor)-aid proteins were used as control strains that produce extended RNA products. The mRNA of M-phase upregulated genes (*ecm33*[+] and *gas1*[+]) was amplified by RT-PCR and separated on a 1.5% agarose gel. Lane numbers indicate reverse primer (Rv1 or 2) used in RT-PCR. M: 100 bp ladder size marker. Although antisense transcripts overlapping with Fw and Rv2 primers could also be detected in this experiment, strand-specific read-through RNAs were verified for *gas1*[+] gene by 3′ RACE (see below). The *nda3 cut14-aid* strain did not produce extended transcriptional products, unlike those in the *nda3 dhp1-aid* or *rna14-aid* strains.

transcribed DNA at mitotically activated genes, although condensin does not impede transcriptional activation and termination.

## 2.5. Condensin mutants show synthetic growth defects with mutants in RNA processing factors

We found negative genetic interactions between condensin mutants and RNA processing factor mutants. Double mutants involving condensin ts mutants (*cut3-477* or *cut14-208*) and RNA processing factor ts mutants (*rna14-393* or *dhp1-154*) were unable to form colonies at the semi-permissive temperature (33°C), although the single mutants grew at this temperature (figure 4*a*). The *cut14-208* ts mutant showed synthetic lethality with the *dis3-54* cs mutant, which is defective in post-transcriptional RNA cleavage at the 3′-end (figure 4*b*) [40]. The Cut14 hinge ts mutant, *cut14-y1*, formed tiny colonies in combination with *cft1-665* and *rna14-393* ts mutants even at the permissive temperature

(26°C) (figure 4*c*). The *seb1-41* ts mutant of RNA-binding and 3′-end-processing protein Seb1 [36,41] was also additively defective in growth with *cut14-208* and *cut14-y1* ts mutants (figure 4*d*). Although the degree of additive effect varies among mutant alleles, failure of mRNA processing at termination sites may require the assistance of condensin in segregating mitotic chromosomes.

## 3. Discussion

In this study, we first constructed auxin-inducible degron strains of condensin or RNA-processing factors, in combination with the *nda3* β-tubulin cs mutant. The aid-*nda3* pair is a unique reagent, capable of inactivating the target protein during mitotic arrest, without releasing cells to anaphase, because the *nda3* cs mutation produces an extremely high rate of synchronization (70–80%) in early mitosis (prophase-like stage) in *S. pombe*. Using the aid-*nda3* strains, we next demonstrated that RNA processing at transcriptional

royalsocietypublishing.org/journal/rsob *Open Biol.* **9**: 190125

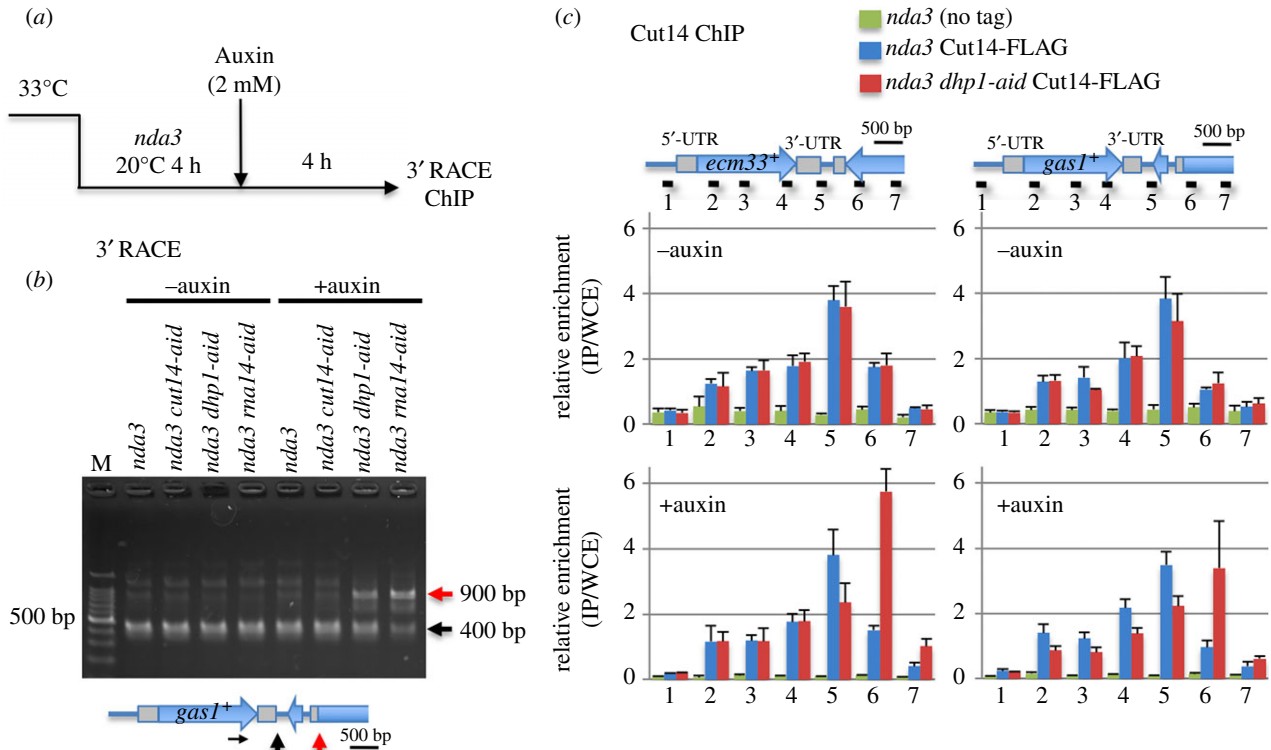

**Figure 3.** Condensin accumulation sites at the 3′-UTR are shifted 500 bp downstream by destruction of 5′-3′ exoribonuclease Dhp1. (*a*) Procedures of Cut14 degradation in mitotically arrested cells for 3′ RACE (rapid amplification of cDNA ends) and ChIP (chromatin immunoprecipitation) assays. (*b*) Transcriptional termination sites at the *gas1*+ gene were determined by 3′ RACE in *nda3* cells expressing Cut14-aid, Dhp1-aid or Rna14-aid proteins. Electrophoretic separation of PCR products in the presence or absence of auxin (top) presented the canonical termination (400 bp, black arrow) and abnormal termination (900 bp, red arrow). M: 100-bp ladder size marker. A schematic (bottom) indicates the position of the forward primer (horizontal black arrow), the canonical termination (black vertical arrow) and the abnormal termination (red vertical arrow) sites. (*c*) ChIP-qPCR analysis of Cut14-FLAG was performed at the two genes (*ecm33*+ and *gas1*+) in single *nda3* and double *nda3 dhp1-aid* mutants along with a non-tagged control strain. Cells were mitotically arrested in the absence (Dhp1 is intact) or presence (Dhp1 is degraded) of 2 mM auxin. Positions of PCR primers (horizontal lines with numbers) are indicated. Relative enrichment was calculated as the ratio of IP to WCE with error bars showing standard deviation (*n* = 3). Association sites of Cut14 at the mitotically activated *ecm33*+ and *gas1*+ genes were shifted further in the 3′ direction by destruction of Dhp1.

termination defines condensin enrichment sites at mitotically activated genes, although condensin deficiency affects neither transcriptional activation nor termination. Our results reveal the selective accumulation of condensin at DNA regions in which the 3′-end of transcribed RNA should be processed. Condensin's action is presumably required especially under conditions in which processing of transcribed RNA and subsequent transcriptional termination are impacted, such as demonstrated in *dhp1-aid* strains. Additive growth defects between condensin ts and RNA processing mutants suggest that abnormally extended termination blocks mitotic progression in the absence of condensin activity. Although the mechanism by which improper RNA processing affects mitosis remains to be addressed, our finding supports the possibility that condensin acts on removal of mitotic transcripts from chromosome DNA for proper chromosome segregation (figure 5).

In *S. pombe*, several RNA processing factors have been identified as negative regulators of condensin-mediated chromosome condensation. A component of the Cleavage and Polyadenylation Factor (CPF), Swd2.2, and an ATP-dependent DNA/RNA helicase, Sen1, antagonize RNAP III-dependent transcription and condensin localization at RNAP III-transcribed genes [42,43]. This seems to contrast with the additive effects of condensin and RNA processing

factors, such as Dhp1 and Rna14, in this study. We speculate that different kinds of RNA processing factors may affect condensin binding at actively transcribed genes in distinct ways, depending on cell-cycle stages, RNA polymerases, and so on. Further study is needed to elucidate specific molecular links between condensin and each factor.

How does condensin accumulate at transcriptional termination sites? Considering the shift of condensin accumulation to 3′ downstream in *dhp1-aid* cells, condensin may prefer specific nucleotide structures or proteins that accumulate at the shifted termination sites. Condensin binds to RNAP II-driven active genes in a transcription-dependent manner in fission yeast and human cells [12,13]. It is also known that *S. pombe* RNAP II enriches at 3′-ends of highly transcribed genes [13,44,45], suggesting that condensin binds in proximity to pausing sites of RNAP II. Transcription by RNAP II accompanies unwinding of dsDNA, and forms DNA-RNA hybrids plus the displaced ssDNA (known as the R-loop structure) [46]. Actually, the *S. pombe* condensin SMC dimer and its holo-complex preferentially bind to ssDNA rather than dsDNA *in vitro* [26]. Sutani *et al.* [13] reported that ssDNA was enriched in Cut14-coprecipitated DNA *in vivo*. SMC dimer and holo-condensin promote the DNA renaturation reaction, which winds up complementary ssDNA to dsDNA *in vitro* [39]. In

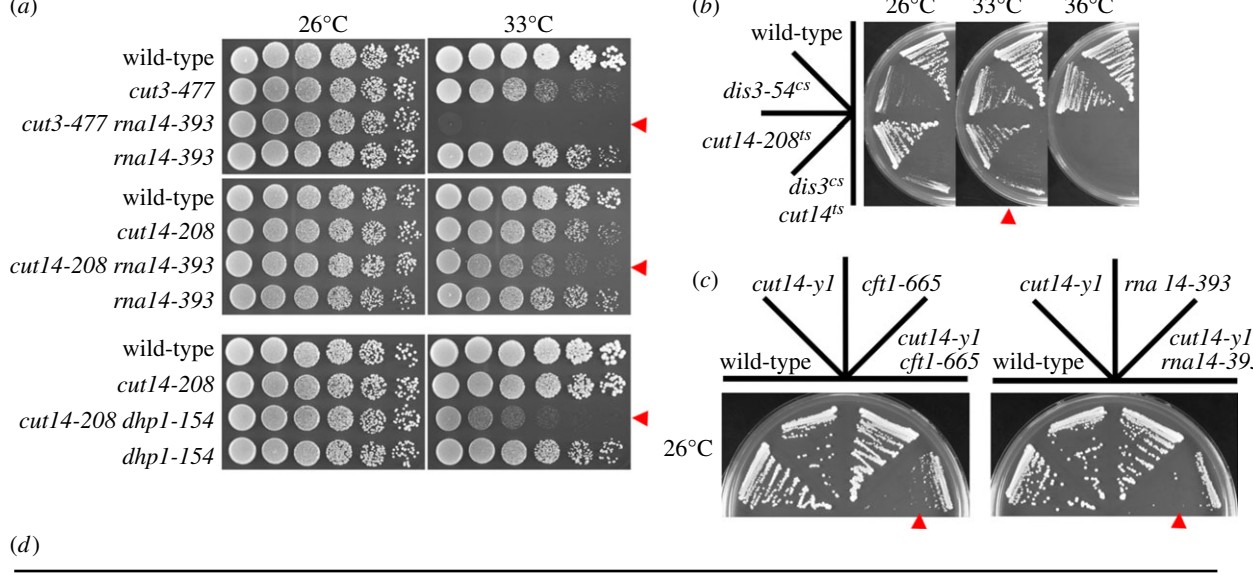

**Figure 4.** Genetic interactions of condensin with 3′-end RNA-processing factors. (*a*) Additive ts phenotype of double mutants between *rna14-393* or *dhp1-154* and *cut3-477* or *cut14-208* at 33℃ (arrowheads). (*b*) Additive lethality between cs *dis3-54* (mutation of 3′-5′ exoribonuclease) and *cut14-208* condensin mutants at 33℃. (*c*) A synthetic growth defect between *cut14-y1* hinge mutant and mutants defective in mRNA cleavage and polyadenylation (*cft1-665* and *rna14-393*) at 26℃. (*d*) Summary of genetic interaction between three condensin mutants and five 3′-end RNA-processing factor mutants.

| mutant strains defective in transcriptional termination | smc2/cut14-208^ts (coilded-coil) | smc2/cut14-y1^ts (hinge) | smc4/cut3-477^ts (coilded-coil) |
|---|---|---|---|
| *rna14-393*^ts (mRNA cleavage and polyadenylation factor) | SD | SD | SD |
| *cft1-665*^ts (mRNA cleavage and polyadenylation factor) | – | SD | – |
| *dhp1-154*^ts (5′-3′ exoribonuclease) | SD | SD | – |
| *dis3-54*^cs (exosome 3′-5′ exoribonuclease ) | SD | not-tested | not-tested |
| *seb1-41*^ts (RNA-binding and 3′-end processing protein) | SD | SD | – |

SD: synthetic growth defect, – : no additive effect

addition, a ssDNA-binding protein, Ssb1/RPA, bound to ssDNA, and also RNA molecules hybridized to ssDNA, were removed by condensin SMC [26]. Therefore, we speculate that condensin recognizes unwound ssDNA, which forms just behind RNAP II. Alternatively, RNAP II-mediated transcription may shift condensin association sites from their initial recruitment sites on DNA to transcriptional termination sites. Condensin is presumably required for renaturation of complementary ssDNA to dsDNA at transcriptional termination sites.

At RNAP III-transcribed gene loci, *S. pombe* condensin enriches at DNA regions accumulating DNA-RNA hybrids, but they are unlikely to facilitate condensin recruitment [43]. It would also be worth testing whether DNA-RNA hybrids directly recruit condensin at RNAP II-transcribed genes in mitosis.

The physiological significance of condensin accumulation at actively transcribed regions remains to be elucidated. By visualizing sister chromatid DNA separation, we recently showed that mutant condensin causes delayed segregation specifically at the mitotically transcribed, condensin-bound gene loci, *ecm33*[+] [47]. Contrarily, the delay was abolished by transcriptional shut-off of the actively transcribed gene. Transcriptional attenuation using transcriptional inhibitors or mutations of the transcriptional mediator protein suppresses chromosome segregation defects in condensin mutants [13,47]. Taken together, we hypothesize that condensin-mediated elimination of RNA transcripts from transcriptional termination sites facilitates mitotic chromosome segregation. Indeed, many genes are actively transcribed even during mitosis, including *ecm33*[+] and *gas1*[+] genes [48,49]. Condensin action may make mitotic transcription compatible with chromosome segregation, probably by liberating the obstructive transcripts immediately after transcription.

# 4. Material and methods

## 4.1. Strains and media

The *S. pombe* haploid wild-type strain 972 *h*[−] and its derivative mutant strains, including the temperature-sensitive (ts) *cut14-208*, *cut14-y1* and *cut3-477*, and cold-sensitive (cs) *dis3-54* and *nda3-KM311*, were used [2,26,28,40]. Strains with chromosomally integrated 3FLAG-tagged Cut14 were described previously [12]. The *rna14-393*, *dhp1-154*, *cft1-665*, and *seb1-41* ts mutants were isolated from a library of 1015 ts mutants [50]. The *dhp1*[+], *rna14*[+], and *cut14*[+] genes were C-terminally tagged with HA and the target peptide of auxin, IAA17, and then integrated into strain FY21102, expressing the Skp1-AtTIR1-NLS-9myc fusion protein, as described previously [30]. The FY21102 strain was provided by the National Bio-Resource Project (Japan Agency for Medical Research and Development, Japan). A synthetic auxin, 1-naphthaleneacetic acid (Nacalai Tesque) was added

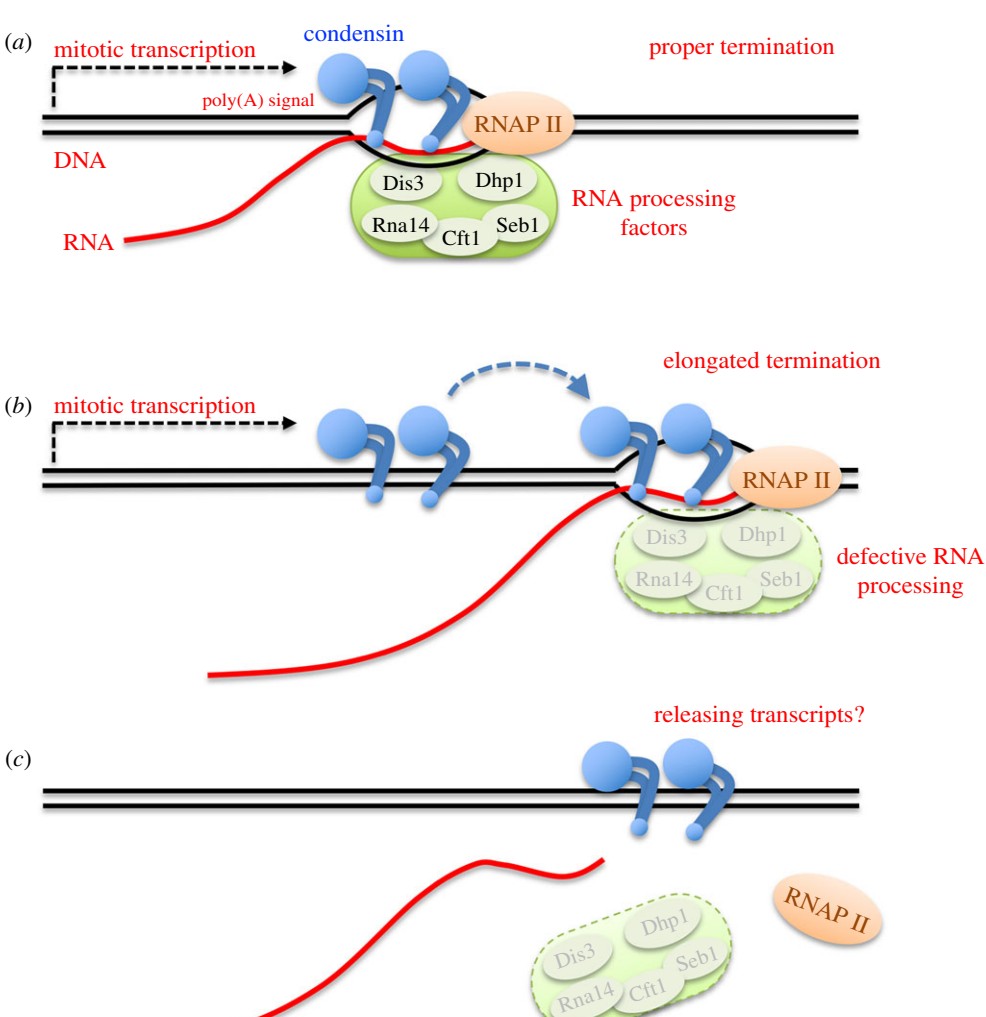

**Figure 5.** Possible action of condensin at transcriptional termination sites for releasing mitotic transcripts. (*a*) Condensin accumulates at transcriptional termination sites at mitotically activated genes. (*b*) When processing of transcribed RNA is obstructed, condensin recognizes the elongated 3′-end of transcribed DNA. (*c*) Considering *in vitro* activities of *S. pombe* condensin, it may remove obstructive mitotic transcripts from DNA for proper chromosome segregation. Failure of RNA processing at termination sites requires condensin activity in segregating mitotic chromosomes.

at the indicated concentration (diluted with dimethyl sulph-oxide) to YPD medium. Culture media used for *S. pombe* were YPD and SPA sporulation medium [2]. Cells were counted using the Multisizer 3 (Beckman Coulter).

## 4.2. Synchronous culture and temperature-shift experiments

Temperature-shift experiments using *nda3-KM311* mutant were performed as previously reported [51]. When cell concentrations reached $4 \times 10^6$ cells ml$^{-1}$ at 30°C (a permissive temperature), cultures were transferred to 20°C and incubated for 4 h to block spindle formation and to arrest cells at an early mitotic stage [28]. For auxin-induced destruction of target proteins, auxin was added to the medium after cells were shifted to 20°C for 4 h, and then incubated for an additional 4 h.

## 4.3. Chromatin immunoprecipitation

ChIP was performed as previously described [12]. Precipi-tated DNA was used as a template and quantified with real-time PCR (Exicycler, Bioneer) using SYBR premix Ex Taq II solution (TaKaRa, RR820). PCR primer sequences are available upon request.

## 4.4. RNA extraction, reverse transcription PCR and 3′RACE analysis

Total RNA from *S. pombe* cells was extracted using the hot-phenol method [52]. One μg of total RNA was reverse-transcribed using PrimeScript RT reagent (TaKaRa, RR047) with oligo dT primers. A genomic DNA eraser supplied with reverse transcription reagent was used to remove con-taminating genomic DNA from the RNA sample. cDNA solution was diluted 25-fold with RNase-free water (Ambion) and 5 μl were used for PCR. Results were quantified using real-time PCR with SYBR premix Ex Taq II solution (TaKaRa, RR820). For 3′RACE analysis, total RNA was extracted as above and reverse transcribed with an adaptor-containing oligo dT primer using a 3′-Full RACE core set (TaKaRa, 6121). cDNA was amplified using Ex Taq polymer-ase (TaKaRa, RR001) with primers in gene ORF and in oligo dT adaptors. Amplified DNA fragments were separated by agarose gel electrophoresis. PCR primer sequences are available upon request.

## 4.5. Northern blotting

A 1.2% agarose gel containing 6.3% formaldehyde in MOPS buffer was prepared and used for electrophoresis. 30 μg of

total RNA was loaded into each lane. For probe preparation, PCR fragments of target genes were amplified from *S. pombe* genomic DNA. Purified PCR fragments were labelled with Alkphos direct labelling (GE Healthcare, RPN3680) and detected with CDP-*Star* chemiluminescent detection reagents (GE Healthcare, RPN3682) using the LAS3000 imaging analyser (Fujifilm).

## 4.6. Immunochemistry

Protein extracts were prepared by TCA precipitation [53]. Cell cultures were harvested by adding 1/4 volume of TCA (final concentration 20%). Cell extracts were prepared using glass beads in 10% TCA and the pellet was boiled with LDS sample buffer (Invitrogen). Immunoblotting was performed using the following antibodies: anti-FLAG M2 (Sigma, Cat# F3165, RRID:AB 259529), anti-HA (Roche, Cat# 11666606001, RRID:AB 514506), anti-tubulin TAT1 (a gift from Dr Keith Gull, University of Oxford, UK).

## 4.7. Fluorescence microscopy

DAPI-staining was carried out as previously described [54]. All-in-one microscopes, BZ9000 and BZ-X700 (Keyence, Japan), were used to observe glutaraldehyde-fixed cells.

Data accessibility. Supplementary figures are available as electronic supplementary material.

Authors' contributions. N.N. and M.Y. designed research. N.N. and O.A. performed experiments. N.N. analysed data and N.N. and M.Y. wrote the paper.

Competing interests. We declare we have no competing interests.

Funding. We received no funding for this study.

Acknowledgements. We are indebted to Dr Shigeaki Saitoh for technical advice, to Dr Kenichi Sajiki and the National Bio-Resource Project (Japan Agency for Medical Research and Development, Japan) for the *S. pombe* strains and to Dr Keith Gull for anti-tubulin TAT1 antibody. We thank Dr Steven D. Aird for editing the manuscript. We are also grateful for the generous support of OIST.

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
