## [Reviewer comments · Open Biology]

Review History

RSOB-19-0125.R0 (Original submission)

Review form: Reviewer 1 (Pascal Bernard)

Recommendation

Major revision is needed (please make suggestions in comments)

Do you have any ethical concerns with this paper?

No

Comments to the Author

Review in the attached file

Review form: Reviewer 2

Recommendation

Accept with minor revision (please list in comments)

Do you have any ethical concerns with this paper?

No

Comments to the Author

In order to evaluate the role of condensin in transcriptional termination, the authors constructed auxin-inducible degron (AID) strain for the condensin Cut14/SMC2 subunit in *S. pombe*. It was demonstrated that transcription initiation is not affected in condensin-AID cells. Moreover, condensin was not found to be required for termination of transcription at mitotically transcribed genes.

The most interesting result is that condensin is accumulated at sites of transcription termination. The authors speculate that condensin acts on removal of mitotic transcripts from chromosome DNA.

However I suggest to discuss alternative mechanism: RNA polymerase shifts condensin and cohesin complexes from transcriptional units leading to their accumulation at sites of transcription termination. By my opinion, the hypothesis that condensin mediates elimination of RNA transcripts from transcriptional termination sites is far-reaching.

I also suggest to change the term "liberating transcripts" to "released transcripts" on fig. 5 and the title of the manuscript.

The manuscript is well-written and is illustrated by nicely made figures and schemes. The overall quality of the paper suitable for publication. I suggest to accept this manuscript after minor revision.

Decision letter (RSOB-19-0125.R0)

15-Jul-2019

Dear Dr Nakazawa

We are pleased to inform you that your manuscript RSOB-19-0125 entitled "Condensin recognizes and locates at transcriptional termination sites in mitosis, possibly liberating mitotic transcripts" has been accepted by the Editor for publication in Open Biology. The reviewer(s) have recommended publication, but also suggest some minor revisions to your manuscript. Therefore, we invite you to respond to the reviewer(s)' comments and revise your manuscript.

Please submit the revised version of your manuscript within 7 days. If you do not think you will be able to meet this date please let us know and we can extend this deadline for you.

- 1) A text file of the manuscript (doc, txt, rtf or tex), including the references, tables (including captions) and figure captions. Please remove any tracked changes from the text before submission. PDF files are not an accepted format for the "Main Document".
- 2) A separate electronic file of each figure (tiff, EPS or print-quality PDF preferred). The format should be produced directly from original creation package, or original software format. Please note that PowerPoint files are not accepted.
- 3) Electronic supplementary material: this should be contained in a separate file from the main text and meet our ESM criteria (see <http://royalsocietypublishing.org/instructions-authors#question5>). All supplementary materials accompanying an accepted article will be treated as in their final form. They will be published alongside the paper on the journal website and posted on the online figshare repository. Files on figshare will be made available approximately one week before the accompanying article so that the supplementary material can be attributed a unique DOI.

Online supplementary material will also carry the title and description provided during submission, so please ensure these are accurate and informative. Note that the Royal Society will not edit or typeset supplementary material and it will be hosted as provided. Please ensure that the supplementary material includes the paper details (authors, title, journal name, article DOI). Your article DOI will be 10.1098/rsob.2016[last 4 digits of e.g. 10.1098/rsob.20160049].

- 4) A media summary: a short non-technical summary (up to 100 words) of the key findings/importance of your manuscript. Please try to write in simple English, avoid jargon, explain the importance of the topic, outline the main implications and describe why this topic is newsworthy.

Images

Data-Sharing

It is a condition of publication that data supporting your paper are made available. Data should be made available either in the electronic supplementary material or through an appropriate repository. Details of how to access data should be included in your paper. Please see <http://royalsocietypublishing.org/site/authors/policy.xhtml#question6> for more details.

Data accessibility section

- DNA sequences: Genbank accessions F234391-F234402

- Phylogenetic data: TreeBASE accession number S9123
- Final DNA sequence assembly uploaded as online supplemental material
- Climate data and MaxEnt input files: Dryad doi:10.5521/dryad.12311

Sincerely,

The Open Biology Team
mailto:openbiology@royalsociety.org

Editor's comment: Please address all comments of reviewers - many of which can be made in revising the discussion section.

Reviewer(s)' Comments to Author:

Referee: 1

Comments to the Author(s)
Review in the attached file

Referee: 2

Comments to the Author(s)

In order to evaluate the role of condensin in transcriptional termination, the authors constructed auxin-inducible degron (AID) strain for the condensin Cut14/SMC2 subunit in *S. pombe*. It was demonstrated that transcription initiation is not affected in condensin-AID cells. Moreover, condensin was not found to be required for termination of transcription at mitotically transcribed genes.

The most interesting result is that condensin is accumulated at sites of transcription termination. The authors speculate that condensin acts on removal of mitotic transcripts from chromosome DNA.

However I suggest to discuss alternative mechanism: RNA polymerase shifts condensin and cohesin complexes from transcriptional units leading to their accumulation at sites of transcription termination. By my opinion, the hypothesis that condensin mediates elimination of RNA transcripts from transcriptional termination sites is far-reaching.

I also suggest to change the term "liberating transcripts" to "released transcripts" on fig. 5 and the title of the manuscript.

The manuscript is well-written and is illustrated by nicely made figures and schemes. The overall quality of the paper suitable for publication. I suggest to accept this manuscript after minor revision.

Author's Response to Decision Letter for (RSOB-19-0125.R0)

See Appendix A.

RSOB-19-0125.R1 (Revision)

Review form: Reviewer 1

Recommendation

Accept with minor revision (please list in comments)

Do you have any ethical concerns with this paper?

No

Comments to the Author

The authors have addressed most of my comments regarding the citations of pre-existing work. However, they performed none of the experiments I requested in order to (1) strengthen the idea that transcription termination positions condensin at the 3' end of actively transcribed genes and (2) to rule out mere indirect genetic interactions between condensin and transcription-termination factors.

In that context I regret to say that I cannot be supportive for a publication in Open biology. I recommend that at least the gene expression analyses requested to complete the data shown in Fig.4 should be performed to consider publication.

Fig. 3. Condensin positioning is changed at altered termination sites

The authors report a shift in the positioning of condensin at the 3'-end of mitotically-transcribed genes upon the depletion of the 5'-3' exoribonuclease Dhp1. Since this shift coincides with the appearance of read-through transcripts, the authors propose that condensin specifically associates with the elongated 3' end of transcribed DNA.

However, a direct alternative conclusion that cannot be ruled out is that Dhp1 itself, or Dhp1-associated proteins, position condensin at the 3' end of Pol-II transcribed genes during mitosis, independently of termination per se.

I requested complementary experiments to investigate this point in my first evaluation. I can understand that the authors want to keep the requested experiments for another project.

However, in that context, I demand that the authors at least clearly mention in the results section that they cannot rule out the possibility that Dhp1 positions condensin independently of the production of read-through transcripts.

Fig. 4. Genetic interactions between condensin and RNA-processing factors

In Fig.4, the authors show negative genetic interactions between condensin and 3'-end RNA processing factors. At first sight these genetic interactions appear consistent with a link between transcription termination and condensin functioning on chromosomes.

However, as I already mentioned an indirect and irrelevant mechanism cannot be ignored in which the transcription termination mutations assessed in Fig. 4 indirectly reduce the expression level of a condensin gene or of the topoisomerase II gene. The authors agreed on that point in their response. Yet, they declined measuring the expression level of condensin sub-units or of the topo II gene, as I requested, and do not even mention this risk in the results section of their manuscript.

Therefore, in the current form of the manuscript, neither the data shown in Fig.3 nor those shown in Fig.4, do unambiguously support the conclusion proposed by the authors.

This is why I reiterate my request that the authors assess the expression level of condensin and topo II genes, at least in the *rna14-393* and *dhp1-154* genetic backgrounds. I think it is important that the authors provide the proof of principle that the genetic interactions they described in Fig.4

are functionally relevant. The requested experiments (RNA extraction and RT-qPCR performed on biological triplicates) are neither labour-intensive, nor time-consuming, and can be easily performed in a 4-6 weeks delay.

Decision letter (RSOB-19-0125.R1)

09-Sep-2019

Dear Dr Nakazawa

We are pleased to inform you that your manuscript RSOB-19-0125.R1 entitled "Condensin locates at transcriptional termination sites in mitosis, possibly releasing mitotic transcripts" has been accepted by the Editor for publication in Open Biology. The reviewer(s) have recommended publication, but also suggest some minor revisions to your manuscript. Therefore, we invite you to respond to the reviewer(s)' comments and revise your manuscript.

Please submit the revised version of your manuscript within 14 days. If you do not think you will be able to meet this date please let us know immediately and we can extend this deadline for you.

- 1) A text file of the manuscript (doc, txt, rtf or tex), including the references, tables (including captions) and figure captions. Please remove any tracked changes from the text before submission. PDF files are not an accepted format for the "Main Document".
- 2) A separate electronic file of each figure (tiff, EPS or print-quality PDF preferred). The format should be produced directly from original creation package, or original software format. Please note that PowerPoint files are not accepted.
- 3) Electronic supplementary material: this should be contained in a separate file from the main text and meet our ESM criteria (see <http://royalsocietypublishing.org/instructions-authors#question5>). All supplementary materials accompanying an accepted article will be treated as in their final form. They will be published alongside the paper on the journal website

and posted on the online figshare repository. Files on figshare will be made available approximately one week before the accompanying article so that the supplementary material can be attributed a unique DOI.

Online supplementary material will also carry the title and description provided during submission, so please ensure these are accurate and informative. Note that the Royal Society will not edit or typeset supplementary material and it will be hosted as provided. Please ensure that the supplementary material includes the paper details (authors, title, journal name, article DOI). Your article DOI will be 10.1098/rsob.2016[last 4 digits of e.g. 10.1098/rsob.20160049].

4) A media summary: a short non-technical summary (up to 100 words) of the key findings/importance of your manuscript. Please try to write in simple English, avoid jargon, explain the importance of the topic, outline the main implications and describe why this topic is newsworthy.

Images

Data-Sharing

It is a condition of publication that data supporting your paper are made available. Data should be made available either in the electronic supplementary material or through an appropriate repository. Details of how to access data should be included in your paper. Please see <http://royalsocietypublishing.org/site/authors/policy.xhtml#question6> for more details.

Data accessibility section

Sincerely,

The Open Biology Team
<mailto:openbiology@royalsociety.org>

Reviewer(s)' Comments to Author:

Referee: 1

Comments to the Author(s)

The authors have addressed most of my comments regarding the citations of pre-existing work. However, they performed none of the experiments I requested in order to (1) strengthen the idea that transcription termination positions condensin at the 3' end of actively transcribed genes and (2) to rule out mere indirect genetic interactions between condensin and transcription-termination factors.

In that context I regret to say that I cannot be supportive for a publication in Open biology. I recommend that at least the gene expression analyses requested to complete the data shown in Fig.4 should be performed to consider publication.

Fig. 3. Condensin positioning is changed at altered termination sites

The authors report a shift in the positioning of condensin at the 3'-end of mitotically-transcribed genes upon the depletion of the 5'-3' exoribonuclease Dhp1. Since this shift coincides with the appearance of read-through transcripts, the authors propose that condensin specifically associates with the elongated 3' end of transcribed DNA.

However, a direct alternative conclusion that cannot be rule out is that Dhp1 itself, or Dhp1-associated proteins, position condensin at the 3' end of Pol-II transcribed genes during mitosis, independently of termination per se.

I requested complementary experiments to investigate this point in my first evaluation. I can understand that the authors want to keep the requested experiments for another project.

However, in that context, I demand that the authors at least clearly mention in the results section that they cannot rule out the possibility that Dhp1 positions condensin independently of the production of read-through transcripts.

Fig. 4. Genetic interactions between condensin and RNA-processing factors

In Fig.4, the authors show negative genetic interactions between condensin and 3'-end RNA processing factors. At first sight these genetic interactions appear consistent with a link between transcription termination and condensin functioning on chromosomes.

However, as I already mentioned an indirect and irrelevant mechanism cannot be ignored in which the transcription termination mutations assessed in Fig. 4 indirectly reduce the expression level of a condensin gene or of the topoisomerase II gene. The authors agreed on that point in their response. Yet, they declined measuring the expression level of condensin sub-units or of the topo II gene, as I requested, and do not even mention this risk in the results section of their manuscript.

Therefore, in the current form of the manuscript, neither the data shown in Fig.3 nor those shown in Fig.4, do unambiguously support the conclusion proposed by the authors.

This is why I reiterate my request that the authors assess the expression level of condensin and topo II genes, at least in the *rna14-393* and *dhp1-154* genetic backgrounds. I think it is important that the authors provide the proof of principle that the genetic interactions they described in Fig.4 are functionally relevant. The requested experiments (RNA extraction and RT-qPCR performed on biological triplicates) are neither labour-intensive, nor time-consuming, and can be easily performed in a 4-6 weeks delay.

Decision letter (RSOB-19-0125.R2)

18-Sep-2019

Dear Dr Nakazawa

We are pleased to inform you that your manuscript entitled "Condensin locates at transcriptional termination sites in mitosis, possibly releasing mitotic transcripts" has been accepted by the Editor for publication in Open Biology.

You can expect to receive a proof of your article from our Production office in due course, please

check your spam filter if you do not receive it within the next 10 working days. Please let us know if you are likely to be away from e-mail contact during this time.

Article processing charge

Please note that the article processing charge is immediately payable. A separate email will be sent out shortly to confirm the charge due. The preferred payment method is by credit card; however, other payment options are available.

Sincerely,

The Open Biology Team
mailto:openbiology@royalsociety.org

Appendix A

Referee#1

In most eukaryotes, condensin is found enriched in the vicinity of highly expressed genes, suggesting the existence of links between the localisation of condensin along chromosomes and features associated with gene expression. In this manuscript, Nakazawa et al. investigate if and how the accumulation of condensin at the 3' end of highly expressed genes in fission yeast is linked to the termination of transcription.

First, the authors assessed whether condensin plays a role in transcription initiation or in the termination of transcription during mitosis. They created a conditional, degron allele of the Cut14 subunit of condensin, depleted Cut14 from mitotically-arrested cells and assessed the initiation and the termination of transcription of two reporter genes. The authors observed no change, and concluded that condensin plays a role neither in the initiation nor in the termination of transcription in mitotically-arrested cells. Next, they investigated whether transcription termination might impinge on the positioning of condensin along chromosomes. Using a degron allele of the termination factor Dhp1, they provide evidence that a defect in the termination of transcription triggered in mitotically arrested cells coincides with a displacement of the condensin peak downstream canonical terminators, suggesting that termination might be a positioning feature for condensin. Finally, the authors describe negative genetic interactions between loss of function mutations in condensin and loss of function mutations in termination factors, and argue that these interactions might reflect functional interplays between the termination of transcription and the localisation and/or the functioning of condensin along chromosomes during mitosis.

The new and major finding reported in this manuscript is the displacement of condensin peaks associated with a failure in transcription termination (Fig. 3). This suggest that features associated with transcription termination might position condensin among chromosomes. However, in my opinion, as it stands this piece of data remains too preliminary and needs to be reinforced by additional supporting data. Similarly, whether the genetic interactions between condensin and termination factors depicted in Fig.4 reflect direct (genuine) or merely indirect mechanisms remains unclear, leaving several alternative scenario. Thus, I think this work conveys a potentially interesting observation, but this observation necessitates additional supporting experiments in order to be published in Open-biology.

I would like to add that I am really surprised that in most parts of their manuscript – characterisation of condensin-degron, role played by condensin in transcription, and investigation of the links between condensin and termination factors – Nakazawa et al. have omitted to cite pre-existing studies published in literature. (1) Degron alleles of the Cut14 and Cnd3 sub-units of fission yeast

condensin have been published (Kakui et al. 2017), but are not mentioned. (2) Lack of direct role for condensin in both gene transcription and transcription-termination has been described in fission yeast through transcriptomics studies (Hocquet et al. 2018), and also in budding yeast (Paul MR et al. 2018). (3) Functional interplays between condensin and the transcription termination machinery have been described in fission yeast (Vanoosthuyse et al. 2014; Legros et al. 2014). In contrast to the conclusion reached by the authors, these studies suggest that components of the termination machinery counteract rather than assist condensin. None of these five published studies, directly relevant to the current work, is cited.

I strongly suggest that the authors cite these missing references whenever appropriate in their revised manuscript.

We thank these constructive comments by this referee and regret not citing above important references. Following the referee's suggestion, we now added the five published work as follows.

(1) Kakui et al. 2017 was referred in the Result section (page 5, lane 103).

(2) Hocquet et al. 2018 and Paul MR et al. 2018 were referred and mentioned in the Introduction section (page 4, lane 77). Please see our response to the referee's comment on the Introduction. Hocquet et al. 2018 was also cited in the Result section (page 7, lane 172). Please see below.

(3) Vanoosthuyse et al. 2014 and Legros et al. 2014 were now clearly referred and discussed in the Discussion section (page 10, lane 236, and page 11, lane 277). Please see below.

Major points

Title

The term “recognizes” used in the title appears inappropriate to me as it implies an ability to distinguish a feature. However, this aspect has not been demonstrated in the manuscript. The data presented do not allow to distinguish between a recruitment of condensin at termination sites from the accumulation of condensin (loaded at other places) at termination sites. I suggest to remove the word recognize from the title.

As suggested by the referee, we removed the word ‘recognize’ from the title.

Abstract

Lane 22: “Herein we demonstrate that condensin does not affect transcription itself”. This has already been shown by previous work.

Lane 23: “Instead, RNA processing at transcriptional site defines condensin accumulation sites...”. This appears to me as an overstatement since (1) the effect of RNA processing per se has not been

demonstrated, and (2) many aspects of transcription termination, e.g. RNA Pol II occupancy, have not been investigated in this manuscript and might well be responsible for the positioning of condensin. I suggest that the authors strongly attenuate this sentence.

Lane 31: for the same reasons as mentioned above I suggest to replace the word “recognizes” by “accumulates”.

Following to the referee’s suggestion, we attenuated the sentence at lane 23 of previous manuscript as “Instead, RNA-processing at transcriptional termination appears to define condensin accumulation sites during mitosis, in the fission yeast *Schizosaccharomyces pombe*.” For lane 31 of previous manuscript, we replaced the word “recognizes” by “accumulates”.

Introduction

Lane 43: “condensin is involved in gene regulation”. This aspect is contradictory and several recent publications have concluded that condensin plays no direct role in gene regulation. I suggest that the authors mention the work performed by other labs whenever appropriate (Swygert SG et al. 2019; Hocquet H et al. 2018; Paul MR et al. 2018).

Lane 66 “Hence, the functional relationship between condensin and transcriptional termination of mitotically activated gene remains elusive”. Again, this aspect has been investigated in other studies, at least on asynchronous cycling cell populations.

According to the referee’s comment, we deleted “gene regulation” from the sentence at lane 43 of previous manuscript. We also referred three published works in Introduction section and mentioned as follows (page 4, line 77 in the revised manuscript).

“At least in asynchronous cycling cell populations, a direct role for condensin in gene regulation has been denied in fission yeast (Hocquet et al., 2018). Budding yeast condensin showed contradictory roles of condensin in transcription under cycling and quiescent conditions (Paul MR et al., 2018; Swygert et al., 2019).”

Results

Results related to Fig1. Characterisation of the cut14-aid mutant strain

- The authors provide evidence that Cut14 is reduced to 10% of the initial amount upon adjunction of auxin 2mM in the cut14-aid genetic background (Fig. 1B). In the presence of auxin (2mM), cut14-aid cells exhibit chromosome segregation defects and > 60% of cells exhibit chromosomes that are cut by the septum upon mitotic exit (Fig. 1E). Yet the cut14-aid strain remains viable and forms colonies on plates containing auxin 2mM (Fig. 1C). Can the authors explain how a fission yeast mutant strain that

experiences > 60% of chromosome cutting upon mitotic exit remain viable?

- Second, since the cut14 gene is essential for growth, the survival of the cut14-aid strain in the presence of auxin 2 mM implies that condensin is not fully inactivated in this condition. This should be mentioned in the text.

- Finally, the thermosensitive (ts) cut14-208 allele used by the authors in their prior studies prevents cell growth at the restrictive temperature of 36°C. Does this imply that the cut14-aid allele in the presence of auxin used here remains less penetrant than the ts cut14-208 allele at 36°C? If yes, the authors should take this limitation into account when interpreting their results.

We agree that the *cut14-aid* strain partly forms colonies on plates containing 2mM auxin (Fig. 1C), although more than 60 % cells exhibit chromosome segregation defect showing a lethal ‘cut’ phenotype after 6 hr at 36 °C (Fig. 1DE). We infer that effect of auxin addition on cell division varies between solid plates and liquid media. According to the microscopic analysis in liquid cultural media, the *cut14-aid* cells presented cut phenotype to similar extent as observed in *cut14-208* at 36 °C. To avoid misinterpretation, we added a phrase, “at least in liquid culture condition”, in the revised manuscript (page 5, lane 116).

Fig. 2 – related to the role of condensin in transcription and termination of transcription

Lane 103. “To fully inactivate condensin... we used the cut14-aid degron”. Since the results shown in Fig.1 indicate that cut14-aid does not allow to fully inactivate condensin, I suggest that this sentence is changed.

As this referee mentioned, we agree that condensin may not be fully inactivated using the *cut14-aid* strain. However, main purpose of construction of this strain is to inactivate condensin in mitotically arrested cells, without releasing cells to anaphase. We now changed this sentence as follows. “To sufficiently inactivate condensin in order to examine whether defective condensin affects transcriptional induction in mitotically arrested cells...” (page 6, lane 133)

Figure 2D-E. The assay used to assess the production of read-through transcripts is not stranded, and, therefore, does not allow to unambiguously identify extended RNAs as read-through RNAs. Total RNA were reverse-transcribed by oligo-dT and cDNAs amplified with primers Fw/Rv1 or Fw/Rv2 (see Fig. 2D). With this setting, antisense transcripts overlapping with the Fw and Rv2 primers could be reverse-transcribed into cDNA, PCR-amplified and inappropriately categorised as read-throughs. Thus, to demonstrate that the extended RT-PCR products do correspond to read-through RNAs, the author should perform strand-specific RT (such as RACE-PCR shown in Fig. 4 for the gas1 gene) As a consequence, the statement lanes 130 and 131 “500bp products....represent abnormally

extended transcripts” appears incorrect to me. The authors must, either perform strand-specific RT or mention in the text the limitation of their experiment and rephrase their presentation.

We explained a possibility that this method detects antisense transcripts overlapping with the Fw and Rv2 primers in the figure legend for Fig. 2 as follows (page 23, lane 625). “Although antisense transcripts overlapping with Fw and Rv2 primers could also be detected in this experiment, strand-specific read-through RNAs were verified for *gas1*⁺ gene by 3’ RACE (see below).”. We also toned down the statement as follows (page 7, lane 164). “In addition to the ~300bp RT-PCR products (Fw-Rv1), ~500bp products (Fw-Rv2), which appear to represent abnormally extended transcripts (Fig. 2D),,,”.

Lane 137 and 138. The authors conclude that their data indicate that condensin is not required for transcriptional termination at mitotically transcribed genes. Perhaps the authors should mention that their observations corroborate previous transcriptomic analyses performed on fission yeast condensin mutants (Hocquet et al. 2018).

We mentioned as follows (page 7, lane 172). “This result is consistent with previous transcriptomic analysis performed on asynchronously cycling condensin mutant cells (Hocquet, et al., 2018).”

Fig. 3. Condensin positioning is changed at altered termination sites

*This is a very interesting observation that deserve to be strengthened by additional supporting data. First, it would important to confirm the change in condensin position observed by ChIP in the *rna14-aid* mutant background, to rule out the possibility that *Dhp1* itself positions condensin at the 3’ end of genes.*

*Second, it is expected that transcription termination will affect condensin positioning in –cis and therefore that condensin positioning remains unchanged in *dhp1-aid* or *rna14-aid* mutant cells at genes whose transcription termination is not modified in the absence of *Dhp1* or *Rna14*, e.g. at specific Pol II- or at Pol III-transcribed genes. This must be verified.*

Third, to further assess the relationship between condensin positioning and transcription termination per se, it is important to compare the shift in position of condensin with the expected shift in occupancy of RNA Pol II.

We appreciate this comment. We understand that additional data is required to support this result and to reveal molecular insights into the changes of condensin positioning in *dhp1-aid* cells. However, there are a lot of questions to be addressed, including above important points raised by the referee, and it is difficult to complete them in several months. Thus, we would like to focus on current

results in Fig. 3 as initial report, and we plan to perform a bunch of additional experiments for next work in the near future.

Fig. 4. Genetic interactions with transcription-termination and RNA processing mutations

The genetic interactions described in this part are consistent with a link between transcription termination and condensin localisation and/or functioning. However, an indirect and irrelevant mechanism cannot be ignored in which transcription termination mutations indirectly reduce the expression level of condensin genes or of the topoisomerase II gene, e.g. by transcriptional interference. Loss of function mutation top2-250 is synthetically lethal with cut3-477 at the permissive temperature of 30°C (Saka et al. 1994).

Thus, to rule out this obvious possibility, RNA levels of condensin sub-units and of topo II must be measured in mutant backgrounds showing synthetic growth defect with condensin.

We agree the point that synthetic growth defect implies a link between condensin and RNA processing factors, but it may include indirect mechanisms. As pointed by the referee, expression level of condensin subunits and/or Top2 proteins might be affected in the mutant strains defective in RNA processing. Unfortunately, we could not rule out this possibility in this work, we are willing to consider it as an next agenda.

Discussion

I suggest that the authors comment on the previous findings made in fission yeast that a component of the Cleavage and Polyadenylation Factor (CPF) and the termination factor Sen1 both hinder the localisation of condensin along chromosomes and act as a negative regulator of condensin-mediated chromosome condensation (Vanoosthuysse et al. 1994; Legros et al. 1994).

Following the referee's suggestion, in the Discussion section we have commented on the previous findings in Vanoosthuysse et al. 2014 and Legros et al. 2014, as follows (page 10, line 236).

“In *S. pombe*, several RNA processing factors have been identified as negative regulators of condensin-mediated chromosome condensation. A component of the Cleavage and Polyadenylation Factor (CPF), Swd2.2, and an ATP-dependent DNA/RNA helicase, Sen1, antagonize RNAP III-dependent transcription and condensin localization at RNAP III-transcribed genes (Vanoosthuysse et al. 2014; Legros et al. 2014). This seems to contrast with the additive effects of condensin and RNA processing factors, such as Dhp1 and Rna14, in this study. We speculate that different kinds of RNA processing factors may affect condensin binding at actively transcribed genes in distinct ways, depending on cell-cycle stages, RNA polymerases, and so on. Further study is needed to elucidate specific molecular links between condensin and each factor.”

Based on existing data and their results, Nakazawa et al. speculate that condensin might recognise features associated with DNA:RNA hybrids formed at transcribed genes (e.g. ssDNA). However, the authors neglect to mention that a role for DNA:RNA hybrids in the localisation of condensin has already been envisaged and experimentally assessed by Legros et al. 1994. Using two independent mapping techniques, Legros et al. have provided evidence that DNA:RNA hybrids are unlikely to facilitate the recruitment of condensin in fission yeast. It would be fair to mention this piece of work in the discussion.

We clearly mentioned as follows (page 11, line 277). “At RNAP III-transcribed gene loci, *S. pombe* condensin enriches at DNA regions accumulating DNA-RNA hybrids, but they are unlikely to facilitate condensin recruitment (Legros et al., 2014). It would also be worth testing whether DNA-RNA hybrids directly recruit condensin at RNAP II-transcribed genes in mitosis.”

Minor points

In Fig. 3, the authors present Condensin-ChIP results as relative enrichments. But do not mention relative to what?

Relative enrichment was calculated as the ratio of IP to WCE (Whole cell extract, equivalent to Input). It is mentioned in the legend of Figure 3.

Referee#2

In order to evaluate the role of condensin in transcriptional termination, the authors constructed auxin-inducible degraon (AID) strain for the condensin Cut14/SMC2 subunit in S. pombe. It was demonstrated that transcription initiation is not affected in condensin-AID cells. Moreover, condensin was not found to be required for termination of transcription at mitotically transcribed genes. The most interesting result is that condensin is accumulated at sites of transcription termination. The authors speculate that condensin acts on removal of mitotic transcripts from chromosome DNA. However I suggest to discuss alternative mechanism: RNA polymerase shifts condensin and cohesin complexes from transcriptional units leading to their accumulation at sites of transcription termination. By my opinion, the hypothesis that condensin mediates elimination of RNA transcripts from transcriptional termination sites is far-reaching.

As suggested by the referee, we discussed an alternative possibility that condensin is pushed by RNAP II-mediated transcription, leading to their accumulation at the transcriptional termination sites, as follows (page 11, line 272).

“Alternatively, RNAP II-mediated transcription may shift condensin association sites from their initial recruitment sites on DNA to transcriptional termination sites.”

I also suggest to change the term "liberating transcripts" to "released transcripts" on fig. 5 and the title of the manuscript.

Following the referee's suggestion, we changed the term “liberating” to “releasing” on Figure 5 and the title.

The manuscript is well-written and is illustrated by nicely made figures and schemes. The overall quality of the paper suitable for publication. I suggest to accept this manuscript after minor revision.

We really appreciate these comments by this referee.